# Intestinal Dysbiosis and Immune Activation in Kawasaki Disease and Multisystem Inflammatory Syndrome in Children: A Comparative Review of Mechanisms and Clinical Manifestations

**DOI:** 10.3390/biomedicines13081953

**Published:** 2025-08-10

**Authors:** Julia Soczyńska, Ewa Topola, Wiktor Gawełczyk, Szymon Viscardi, Kamila Butyńska, Sławomir Woźniak

**Affiliations:** 1Student Scientific Society Anatomia-Klinika Nauka, Division of Anatomy, Department of Human Morphology and Embryology, Wroclaw Medical University, 50-367 Wroclaw, Poland; ewa.topola@student.umw.edu.pl (E.T.); wiktor.gawelczyk@student.umw.edu.pl (W.G.); szymon.viscardi@student.umw.edu.pl (S.V.); kamila.butynska@student.umw.edu.pl (K.B.); 2Division of Anatomy, Department of Human Morphology and Embryology, Wroclaw Medical University, 50-367 Wroclaw, Poland; slawomir.wozniak@umw.edu.pl

**Keywords:** Kawasaki disease, pediatric inflammatory multisystem syndrome, multisystem inflammatory syndrome in children, intestinal dysbiosis

## Abstract

Although Kawasaki disease (KD) has been known since 1967, when it was first described by Dr. Tomisaku Kawasaki, the literature indicates that its etiology—similarly to Multisystem Inflammatory Syndrome in Children (MIS-C)—remains largely unclear and is the subject of intensive research. The former disease, which typically occurs shortly after infection, is the most common cause of primary vasculitis in children worldwide. The latter—MIS-C, associated with SARS-CoV-2 infection—is characterized by involvement of at least two organ systems. Undoubtedly, both diseases exhibit heightened immune system activity and significant inflammation. In recent years, increasing attention has been directed towards alterations in the microbiota observed in affected patients. We undertake an analysis and systematic review of the current scientific findings in this field. We emphasize the role of the microbiome—which encompasses not only bacteria but also viruses, fungi, parasites, and archaea—in health and disease. We track its composition from birth and highlight factors influencing its diversity, such as the mode of delivery. We recognize the microbiome’s role in reducing the likelihood of allergic diseases in children and its interactions with the immune system. In addition to comparing the pathomechanisms and clinical manifestations of KD and MIS-C, also known as Pediatric Inflammatory Multisystem Syndrome (PIMS), we investigate microbiota alterations in these conditions and analyze potential applications of microbiome knowledge, for example, in identifying diagnostic markers. We also point out potential directions for future research, such as the use of short-chain fatty acids (SCFAs) in MIS-C and the long-term changes in the gut microbiota associated with these diseases, which remain poorly documented and currently represent significant gaps in knowledge.

## 1. Introduction

KD and MIS-C are serious conditions predominantly affecting children and are characterized by systemic inflammation [1]. MIS-C, also referred to as PIMS, is often described as a “Kawasaki-like” illness due to its similar clinical presentation [2,3]. Both diseases frequently lead to complications, particularly of a cardiovascular nature [1]. KD is an acute systemic vasculitis, whereas MIS-C is a hyperinflammatory condition typically occurring 2 to 6 weeks following SARS-CoV-2 infection [4,5]. The two syndromes share a range of clinical symptoms, including rash, fever, gastrointestinal disturbances, and cardiovascular involvement. However, they differ in epidemiological features, specific clinical manifestations, and diagnostic criteria [5]. Recent studies have suggested that a significantly altered gut microbiome may play a critical role in the pathophysiology of both conditions [6,7]. Motivated by this observation, we aimed to investigate the relationship between the gut microbiome and the pathomechanisms of these diseases by conducting a comprehensive review of the latest scientific evidence. The intestinal microbiota plays a pivotal role in immune system regulation, including the modulation of immune cell activity, coordination of host defense responses to pathogens, and maintenance of immunological homeostasis [8]. Based on this, it is plausible to assume that disruptions in the gut microbiome are crucial in the development of inflammatory diseases. Microbiota imbalances—referred to as dysbiosis—may lead to inappropriate immune responses and the initiation of pathological inflammation [9]. In MIS-C, dysbiosis is characterized by a reduction in microbial diversity and abundance, contributing to intestinal barrier dysfunction [6,10]. Similarly, in KD, the epithelial barrier of the intestine—constituting the body’s first line of defense—is compromised. It is hypothesized that patients with KD exhibit increased intestinal permeability, which may allow for the translocation of pathogens or microbial products into the bloodstream, thereby triggering a cascade of systemic inflammatory responses characteristic of KD [4]. In KD patients, a marked reduction in SCFA production has been observed, associated with decreased levels of *Blautia* species, which may increase susceptibility to the disease. Moreover, individuals with a history of KD exhibit elevated levels of *Ruminococcus gnavus*, a bacterium known to produce pro-inflammatory polysaccharides [7,11,12]. In the case of MIS-C, particular attention has been drawn to the predominance of *Bacteroides*, *Eggerthella*, and *Prevotella* species. These taxa have been implicated in the production of pro-inflammatory cytokines and are believed to contribute to disease pathogenesis [6]. Dysbiosis has also been linked to other immune-mediated disorders, such as inflammatory bowel diseases, prompting growing interest in therapeutic approaches targeting the microbiota [13,14]. The gut microbiome–immune system axis represents a promising avenue for both diagnostic and therapeutic interventions in KD and MIS-C. Restoration of intestinal barrier function, microbial diversity, and microbiota balance may potentially improve clinical outcomes and disease prognosis. This review aims to elucidate the interplay between dysbiosis, immune regulation, and clinical manifestations in both KD and MIS-C, highlighting the potential of microbiota-based strategies in advancing the understanding and treatment of these pediatric inflammatory disorders.

## 2. The Role of the Gut Microbiome in Health and Disease

The human body is colonized by a diverse array of microorganisms. Among the most heavily populated areas are the intestines, which may harbor between 500 and 1000 distinct microbial species [15]. The term microbiota—commonly referred to as the flora—denotes the collection of microorganisms (including bacteria, viruses, fungi, parasites, and archaea) that inhabit the human body. Microbiota constitutes a component of the microbiome, a broader concept that also encompasses the ecological niche, habitat, and the complex interactions between microorganisms and their host [16].

In the early years of life, the composition of the gut microbiota gradually evolves toward a structure characteristic of adulthood. This process is associated with more efficient utilization of intestinal mucins by microorganisms, enhanced cysteine metabolism, and increased fermentation activity [17]. The maternal vaginal microbiota has been shown to influence the initial composition of the infant gut microbiome. In infants born via vaginal delivery, the microbiota initially resembles that of the mother’s vaginal and fecal flora, with a predominance of *Lactobacillus*, *Prevotella*, and *Sneathia* species, and is measurably more diverse than that of infants born via cesarean section, in whom skin-associated bacteria initially predominate. According to studies, these differences tend to diminish by 12 months of age [18,19,20]. A study by Hollister et al. demonstrated that in childhood (between ages 7 and 12), the gut microbiota is enriched with *Bifidobacterium*, *Faecalibacterium*, and members of the *Lachnospiraceae* family [21]. According to Hickman et al., *Bifidobacterium* during the first two years of life is considered a key indicator of a healthy and well-developing microbiota, exerting beneficial effects through its regulatory role. Importantly, the same authors emphasize the role of microbe transfer through older siblings, which appears to reduce the risk of developing allergies. They also note that while breast milk supports the development of gut microbiota, it is not in itself a sufficient microbial source [22]. In healthy adults, the gut microbiota is predominantly composed of bacteria belonging to the phyla *Firmicutes* and *Bacteroidetes*, although certain strains of *Bacteroidetes* have been associated with inflammatory gastrointestinal disorders [21,23]. In addition to these dominant groups, other phyla present in the gut microbiota include *Actinobacteria*, *Fusobacteria*, *Proteobacteria*, *Verrucomicrobia*, and *Cyanobacteria* [24]. At the genus level, a typical healthy gut microbiota consists of approximately 16% *Bacteroides*, 8.7% *Prevotella*, 7.5% *Faecalibacterium*, and 5.5% *Bifidobacterium* [25]. Recent research suggests that the functional capacity of the microbiome, rather than its exact taxonomic composition, is of primary importance, as different microbial species may fulfill similar biological roles [16]. The gut microbiome is a complex ecosystem that plays a pivotal role in regulating the host’s immune responses [26]. Its functions include the maintenance of gut-associated lymphoid tissue (GALT), interactions with macrophages, and the regulation of T regulatory (Treg) lymphocytes—which are essential for preventing autoimmune diseases—partly by promoting their development and the release of interleukin-10 (IL-10) [27,28]. In comparative studies involving mice raised under standard conditions versus germ-free environments, it has been shown that germ-free mice exhibit impaired GALT development. Furthermore, these mice display significantly reduced numbers of Treg lymphocytes (Foxp3+) in the colonic lamina propria, as well as decreased levels of helper T cells Th1 and Th17, in comparison to conventionally colonized controls. These immunological deficits can be reversed through recolonization of the germ-free mice [15].

In the context of infection prevention, the gut microbiota exerts protective effects not only through direct competition with pathogens for nutrients—where the synergistic interaction among multiple species is more critical than the action of any single one—but also via modulation of host defense mechanisms [29,30]. Well-documented mechanisms include the regulation of immunoglobulin A (IgA) production by the gut microbiota [31], as well as the enhancement of lysozyme activity—a key antimicrobial agent—through lactic acid produced during the metabolism of *Lactobacillus* spp. [32]. The gut microbiota also plays a vital role in the absorption and metabolism of nutrients and certain pharmaceuticals. Commensal bacteria are actively involved in the fermentation of indigestible carbohydrates, leading to the production of SCFAs, which contribute, among other effects, to the regulation of satiety-related hormones [33,34]. Notably, SCFAs such as acetate and butyrate strengthen the intestinal barrier and suppress inflammation [35]. Additionally, the gut microbiome is involved in the biosynthesis of essential vitamins—including vitamin K and B-complex vitamins such as folic acid, pantothenic acid, and biotin—as well as in the assimilation of trace elements like iron, calcium, and magnesium [33,36]. Several factors influence the composition of the gut microbiota, including diet, medications, lifestyle, sleep, stress, and host genetics [37,38,39]. Among these, dietary habits exert a particularly strong influence on microbial selection within the gut. Numerous studies support this claim. For instance, one investigation examined the impact of exclusive carnivorous versus plant-based diets on microbiota diversity. In individuals consuming only animal-derived foods, a significant increase was observed in bacteria capable of metabolizing bile acids, alongside a decrease in *Firmicutes*, which are typically responsible for the metabolism of plant-based carbohydrates [40]. It is also important to note that consumption of processed foods has a detrimental effect on gut microbiota composition, whereas the intake of fermented foods is associated with beneficial microbial modulation [38].

The available data underscore the gut microbiota’s high sensitivity to dietary changes, highlighting the potential for effective nutritional interventions in both the prevention and treatment of various diseases. Gut microbiome health can be supported through a range of strategies, including the use of prebiotics—non-digestible dietary compounds that selectively stimulate the growth and activity of beneficial microorganisms [41]. Microbiological interventions also include the use of probiotics and postbiotics. Probiotics are defined as live microorganisms which, when administered in adequate amounts, confer health benefits on the host [42]. In contrast, postbiotics refer to inactivated microorganisms (or their components), with or without their metabolites, that exert beneficial effects on intestinal health—primarily by enhancing gut barrier integrity. These compounds have been shown to modulate immune responses and, consequently, reduce inflammatory processes [43]. According to Smolinska et al., the precise roles and mechanisms of action of prebiotics, probiotics, and postbiotics in supporting host health are currently being increasingly elucidated [44].

The gut microbiota plays a crucial role in maintaining health and may also significantly contribute to the development of various diseases [45]. However, due to the high diversity of microorganisms inhabiting the gut across different stages of life, it remains challenging to clearly define what constitutes a “healthy” microbiota or a state of dysbiosis [46]. This difficulty is further compounded by the lack of standardized methodologies for analyzing and interpreting microbiome data.

## 3. KD

### 3.1. Epidemiology

KD was first described by Dr. Tomisaku Kawasaki in 1967. It typically manifests shortly after a bacterial or viral infection and affects approximately 0.01% of the pediatric population. The incidence peaks in late winter and spring, with a male-to-female ratio of 3:2. While the disease affects children of all racial and ethnic backgrounds, individuals of Asian descent are at the highest risk. Globally, it is the most common form of primary vasculitis in children [47,48,49]. Interestingly, Japan has experienced three epidemic waves of KD—in 1979, 1982, and 1986 [50]. In both Japan and Taiwan, the incidence is 10 to 20 times higher than in the United States. The disease most commonly occurs in children between 6 months and 5 years of age, with significantly fewer cases reported beyond this age range [51]. The recurrence rate of KD is approximately 3.5%. Additionally, a family history appears to increase susceptibility: 0.7% of patients had a parent previously diagnosed with the disease, and 1.4% had an affected sibling [52].

KD is a self-limiting inflammatory condition that can progress to the development of coronary artery aneurysms. If left untreated, aneurysms develop in approximately 15–25% of cases. The disease primarily affects children—most commonly infants and children under five years of age, accounting for 80% of cases. Treatment typically involves the administration of intravenous immunoglobulin (IVIG) and aspirin, ideally within the first 10 days of illness. IVIG therapy has been shown to prevent the formation of coronary aneurysms in up to 95% of patients [49,53,54,55]. Cases that do not fulfill the full diagnostic criteria are referred to as atypical or incomplete KD. However, even in these cases, there is a documented risk of coronary artery involvement [56]. Atypical KD most often occurs in children under one year of age and may present with signs such as renal impairment, pericardial effusion, and abdominal pain. Although these patients do not meet the classical diagnostic criteria, the diagnosis can be supported by additional findings characteristic of KD—such as coronary aneurysms or abnormal blood count parameters. Timely diagnosis is crucial in preventing complications, particularly the development of coronary artery aneurysms [57].

### 3.2. Pathogenesis

The exact pathogenesis of KD remains unclear; however, a genetic component has been widely suggested. Several studies have identified a potential association with the transforming growth factor-beta (TGF-β) signaling pathway, particularly involving SMAD3, TGF-β2, and TGF-βR2 in European populations [48]. Across different ethnic groups, multiple susceptibility genes have been identified, including FAM167A-BLK, CASP3, FCGR2A, CD40, ORAI1, and ITPKC. The frequency and expression of these genes vary between populations. Genomic regions linked to disease susceptibility have also been identified through linkage and association studies. Furthermore, the expression of the FCGR3B gene may influence both the development of coronary artery aneurysms and the patient’s response to IVIG therapy [58,59].

Another proposed mechanism involves the potential role of infectious agents—particularly RNA viruses—as indicated by the presence of cytoplasmic inclusion bodies in bronchial epithelial cells. A bacterial origin has also been considered, including an immune response to bacterial superantigens from *Staphylococcus* or *Streptococcus* species; however, definitive evidence supporting this theory remains lacking. The complexity of KD pathogenesis is further influenced by the host immune response. Laboratory findings have revealed elevated levels of pro-inflammatory cytokines such as IL-1β, IL-6, and tumor necrosis factor-alpha (TNF-α), along with an increased neutrophil count [48]. In their review, Rowley et al. proposed that the etiological agent could be an asymptomatically spreading RNA virus, supported by the lack of response to antibiotic therapy. Studies have also demonstrated an antigen-driven IgA immune process, reinforcing the infectious hypothesis. Inflammatory infiltrates in affected tissues include CD8+ T lymphocytes, along with upregulation of genes involved in interferon signaling and cytotoxic pathways in coronary arteries—further supporting a viral etiology. It is hypothesized that maternal antibodies may provide protection against the disease during the first six months of life. The suspected virus may be present in inclusion bodies found in bronchial epithelial cells (where the so-called KD antigen has been localized) and could be carried into the bloodstream by macrophages, eventually reaching and affecting susceptible tissues. The effectiveness of IVIG therapy may be explained by the fact that most donors have likely experienced a subclinical form of the disease, thereby providing protective antibodies [60].

The disease predominantly targets medium-sized arteries. In post-mortem examinations of patients who died within two weeks of disease onset, infiltration by neutrophils has been observed, whereas beyond two weeks, there is an increasing presence of eosinophils and CD8+ T cells [49]. Macroscopically, the inflammatory process leads to destruction of the elastic layer of the arterial wall, resulting in the formation of aneurysms and thrombi. During the healing and regeneration phase, the intensity of inflammation decreases, giving rise to granulation tissue, followed by fibrosis and subsequent narrowing of the vessel lumen. Myocardial fibrosis and endocardial fibroelastosis have also been reported [51].

However, studies using a murine model have demonstrated that coronary artery lesions can be induced by administration of a *Lactobacillus casei* cell wall extract. The induction of such lesions requires Toll-like receptor 2 and the adaptor protein MyD88, indicating the involvement of the innate immune system. This is further supported by the upregulation of transcription factors and genes associated with innate immune responses [61].

### 3.3. Intestinal Dysbiosis

An intriguing area of investigation is the role of gut microbiota disturbances and their consequences in the development of KD. Similar conclusions were drawn by Kaneko et al., who suggested that dysbiosis may underlie the pathogenesis of KD. This hypothesis is supported by the theory that an imbalance between Th17 and Treg lymphocytes leads to an inappropriate immune response in the disease. The development of both Treg and Th17 cells is regulated by SCFAs, which are produced by the gut microbiota. The aforementioned compounds also inhibit the activation of the NLRP3, a cytoplasmic protein that becomes activated in response to damage-associated molecular patterns derived from host cells and pathogen-associated molecular patterns originating from infectious agents [4,62,63]. NLRP3 may amplify the uncontrolled production of IL-1 beta, which can contribute to the development of inflammatory diseases. Fluctuations in gut microbiota composition are associated with NLRP3 inflammasome activation. Elevated levels of IL-1 beta and NLRP3 inflammasome have been observed during the acute phase of KD [64,65,66]. Notably, levels of butyrate were found to be reduced in stool samples of KD patients who had not received antibiotics at the time of diagnosis [63]. Dietary modulation and the use of probiotic supplementation may help regulate this imbalance [12]. Other studies have analyzed gut flora with respect to the ratio of butyrate-producing to pro-inflammatory bacteria. For instance, *Blautia* species produce butyrate (which is utilized by colonic epithelial cells), while *Ruminococcus gnavus* is considered a pro-inflammatory bacterium. An imbalance in the populations of these bacteria has been observed in KD patients. Mouse model studies have demonstrated a reduction in intestinal epithelial permeability and decreased production of pro-inflammatory cytokines. Moreover, in some KD patients, alterations have been noted in the function of ABC transporters, which play a role in limiting the colonization of lactic acid-producing bacteria. These bacteria contribute to the proper functioning of the intestinal immune system [7]. Another study identified a broader spectrum of SCFA-producing bacteria that were found in reduced abundance in KD patients. These included *Collinsella*, *Dialister*, *Clostridium*, *Roseburia*, *Dorea*, *Lachnospiraceae* incertae sedis, *Lactobacillus*, *Blautia*, *Bacteroides*, *Faecalibacterium prausnitzii*, and *Akkermansia muciniphila*. In contrast, an increased abundance of pro-inflammatory species was observed, including *Helicobacter*, *Ruminococcus gnavus*, *Escherichia*, *Shigella*, *Bifidobacterium*, *Enterococcus, Acinetobacter*, *Lactococcus*, *Staphylococcus* (*S. mitis*, *S. pneumoniae*, *S. oralis*, and *S. sanguinis*), *Neisseria mucosa*, *Veillonella*, *Eubacterium*, *Peptostreptococcus*, *Fusobacterium*, *Butyricimonas*, and *Finegoldia magna* [67]. Additionally, both the acute and subacute phases of KD are associated with shifts in microbiota composition. The subacute phase is characterized by a predominance of *Ruminococcus*, while the acute phase shows an increase in *Streptococcus* spp., including *S. pneumoniae*, *S. pseudopneumoniae*, *S. gordonii*, *S. sanguinis*, and *S. oralis*. Moreover, a study involving patients with liver cirrhosis demonstrated that certain bacteria, including *Streptococcus* spp., invade the gut and may serve as potential microbial signatures in diseases of unknown etiology [68].

### 3.4. Clinical Manifestation

The clinical manifestations of KD include, among others, fever, rash, swelling of the hands and feet, conjunctivitis, and cervical lymphadenopathy [47]. Diagnosis is based on the persistence of fever lasting more than five days, in combination with additional clinical criteria. In some cases, the duration of fever may be shorter (four days) if IVIG is administered and successfully reduces the fever. The principal diagnostic criteria include edema or erythema of the hands and feet, followed by periungual desquamation; cervical lymphadenopathy (at least one lymph node ≥ 1.5 cm in diameter); non-vesicular polymorphous rash, usually localized to the trunk; bilateral non-purulent conjunctival injection; involvement of the oral mucosa, including erythema; fissuring of one or both lips; pharyngeal erythema; and the characteristic “strawberry tongue”. The diagnosis of KD is established when fever is present alongside at least four of the five principal features [69]. Laboratory findings may support the diagnosis, especially in incomplete or atypical cases. These include elevated C-reactive protein (CRP), anemia, elevated transaminases, hypoalbuminemia (≤3.0 g/dL), hyponatremia, thrombocytosis (platelet count > 450,000/μL), leukocytosis (white blood cell count ≥ 15,000/μL), cerebrospinal fluid pleocytosis, and sterile pyuria [61,70]. Non-specific symptoms may also be present, including gastrointestinal disturbances such as vomiting, diarrhea, abdominal pain, and nausea. Musculoskeletal manifestations include arthralgia and arthritis. Cardiovascular complications are common and may present as mitral valve regurgitation, pericardial effusion, myocarditis, and impaired left ventricular function. Perineal desquamation has also been described as a potential early sign of KD. Notably, reactivation of the *Bacillus Calmette–Guérin* vaccination site may occur, which is considered a useful diagnostic clue. Hydrops of the gallbladder is another known feature of the disease [70,71]. Postmortem studies in fatal cases have revealed intravascular thrombi and coronary artery aneurysms, as previously described. Thrombosis is thought to be associated with disturbed blood flow due to decreased endothelial shear stress. In adults, long-term complications may include ischemic heart disease and chronic coronary insufficiency [47,55,72].

It is also worth considering the role of biomarkers in KD that, although not currently used in routine diagnosis, may prove useful in future diagnostic approaches given the unclear etiology of the disease. Elevated levels of pro-inflammatory cytokines such as TNF-α, interferon-gamma (IFN-γ), IL-6, and IL-20, which are associated with Th1 and Th2 immune responses, have been observed. Following administration of IVIG, levels of these cytokines typically decreased—with the exception of TNF-α, which remained elevated in IVIG-resistant cases. Additionally, elevated levels of IL-17 have been reported. N-terminal pro-brain natriuretic peptide (NT-proBNP) has also been investigated in the context of KD. Although not specific to the disease, when analyzed in conjunction with IL-17, it has shown promise in differentiating atypical KD from other febrile infectious diseases in children. This suggests its potential utility as a future diagnostic tool for KD [71]. Rivas et al. conducted studies in mouse models and highlighted the involvement of IL-1β, a key pro-inflammatory cytokine, in the destruction of endothelial cells, which predisposes individuals to the formation of coronary artery aneurysms. IL-1β thus represents a pathogenic pathway potentially targetable in the treatment of KD. The authors further noted that inhibiting this cytokine may also have therapeutic benefits in other autoimmune diseases characterized by the deposition of IgA antibodies [73].

Echocardiography is the primary imaging modality used in the diagnosis and monitoring of coronary artery aneurysms in KD. It is also employed to assess for valvular abnormalities, pericardial effusion, and overall cardiac function. Importantly, some cardiovascular abnormalities may not become apparent until the subacute phase of the disease. Echocardiography is not used to establish the initial diagnosis of KD; however, findings such as ectasia, aneurysms, and increased echogenicity of the coronary artery walls may be observed during the examination [74]. The standard treatment of KD includes IVIG in combination with aspirin. In cases of disease recurrence or resistance to initial therapy, additional treatment with corticosteroids and other immunomodulatory agents may be required to control symptoms. In the acute phase, IVIG is administered at a dose of 2 g/kg body weight as a single infusion, while aspirin is given at 80–100 mg/kg/day, divided into four doses. This regimen is continued until resolution of fever, typically within 48–72 h [70].

## 4. MIS-C

### 4.1. Epidemiology

MIS-C is a clinical entity associated with SARS-CoV-2 infection, the virus responsible for COVID-19. Although pediatric cases constitute a minority of all COVID-19 infections, recent attention has been drawn to a syndrome involving multiorgan involvement that develops several weeks after the acute phase of infection [75]. MIS-C typically manifests 4–6 weeks following the initial infection [76]. The clinical features of MIS-C overlap with those of toxic shock syndrome and KD, though MIS-C remains a relatively new entity, with the first cases reported in April 2020 [77]. Interestingly, epidemiological observations have revealed varying trends in susceptibility: some studies suggest a higher incidence among White children [78], while others report greater prevalence in African American and Latino populations [79]. In most cases, children who developed MIS-C tested negative on RT-PCR for SARS-CoV-2 but were seropositive for SARS-CoV-2 antibodies, suggesting a post-infectious immune-mediated process [80]. Among individuals under 21 years of age, the incidence of MIS-C is estimated at 2 per 100,000, whereas SARS-CoV-2 infection alone occurs at a rate of 322 per 100,000. Researchers have raised concerns that the broad diagnostic criteria for MIS-C may encompass other diseases, highlighting the uncertain etiology of the syndrome [81]. An observational study involving 662 pediatric patients with MIS-C found that 71% required admission to intensive care units, with 11 reported deaths (1.7%). Fever was present in 100% of cases, while abdominal pain and diarrhea occurred in 73.7% (488 patients), and vomiting in 68.3% (452 patients). Extracorporeal membrane oxygenation was necessary in 4.4% of patients (29 individuals) and mechanical ventilation in 22.2% (147 patients). Notably, 54% (581 individuals) showed abnormal echocardiographic findings, with 45.1% of these (262 cases) demonstrating reduced ejection fraction. Hoste et al. reported that 56.3% of patients presented with signs of shock upon admission [77]. Another study analyzing 186 pediatric cases determined the median age of affected children to be 8.3 years, with a median hospital stay of 7 days. Interestingly, 62% of patients were male [82].

### 4.2. Pathogenesis

Similar to KD, the pathogenesis of MIS-C remains poorly understood, and various hypotheses have been proposed. One suggested mechanism involves antigen mimicry, whereby viral antigens are recognized by T cells or antibodies, leading to an autoimmune-like response. This process results in the production of antibodies that target virus-infected cells, the formation of immune complexes, and activation of systemic inflammation [75]. In a study by Consiglio et al., key differences in pathogenesis between MIS-C and KD were highlighted. These include differences in IL-17A expression (associated with hyperinflammation in KD); T-cell subpopulations (MIS-C shows characteristics between KD and acute SARS-CoV-2 infection), such as lower CD4+ T cell counts; reduced follicular helper T cells; and an increased presence of effector memory and central memory T cells in MIS-C compared to KD. Furthermore, biomarkers related to arterial inflammation were found at higher levels in KD than in MIS-C. The researchers also observed that immunoglobulins bind to immune signaling proteins, as well as to structural proteins in the heart and blood vessels, which may represent future therapeutic targets for MIS-C [76]. Additional studies identified elevated levels of spike protein in circulation and increased expression of matrisome-related genes. MIS-C is associated with immune signatures driven by type II IFN-γ and NF-κB signaling pathways, which appear independent of SARS-CoV-2 PCR test results. A clonal expansion of T cells expressing TRBV11-2 has also been reported, correlating with heightened inflammatory states. A potential genetic predisposition has been proposed, with associations between MIS-C and certain HLA haplotypes, including HLA-A02, B35, and C*04. Additionally, B cells from MIS-C patients exhibit greater genetic alterations, suggesting a more robust immune response to SARS-CoV-2 [83]. The spike protein itself may act as a superantigen, potentially triggering a cytokine storm. Levels of IL-15 are elevated, and patients commonly possess IgG antibodies against SARS-CoV-2. As in KD, MIS-C is associated with increased activity of IL-1β+ cells and immature neutrophils. Furthermore, patients often display features resembling macrophage activation syndrome, characterized by profound systemic inflammation [79].

### 4.3. Intestinal Dysbiosis

A study by Yeoh et al. investigating the gut microbiota composition in COVID-19 patients suggests a potential role of persistently dysbiotic intestinal flora in the development of multisystem inflammatory syndromes, even after the clearance of SARS-CoV-2 [84]. The analysis was conducted using 16S rRNA gene sequencing. In patients diagnosed with MIS-C, an increased abundance of *Bacteroidetes* and a decreased presence of *Firmicutes* was observed when compared to control subjects. Interestingly, the microbiota composition differed at the phylum and species levels when comparing MIS-C patients to those with acute COVID-19. Species such as *Bacteroides uniformis*, *Bacteroides plebeius*, *Clostridium ramosum*, *Prevotella tannerae*, *Bacteroides coprophilus*, *Eubacterium dolichum*, *Eggerthella lenta*, and *Bacillus thermoamylovorans* were more abundant in MIS-C patients, while the presence of *Faecalibacterium prausnitzii*—a species typically associated with gut health and anti-inflammatory effects—was significantly reduced. Notably, an increase in *Dorea formicigenerans* and *Bifidobacterium adolescentis* was observed in COVID-19 patients without MIS-C. Moreover, respiratory illnesses such as COVID-19, particularly those affecting the lungs, are associated with decreases in *Firmicutes* and increases in *Bacteroidetes*. However, the authors did not establish a causal relationship between intestinal microbiota and MIS-C pathogenesis [6]. Another notable finding was that the inflammatory process in MIS-C may lead to antibody production against both cardiovascular and gastrointestinal tissues [85]. A review study proposed that intestinal barrier disruption due to inflammation might contribute to immune dysregulation and that dietary interventions and probiotics could potentially play a therapeutic role. The cytokine IL-6, which exhibits both pro- and anti-inflammatory properties, is elevated in MIS-C. High IL-6 levels are known to compromise gut barrier integrity, enhancing the release of zonulin, a protein that increases intestinal permeability. This process may facilitate the translocation of spike protein and microbial toxins from the gut into systemic circulation. It has also been observed that SARS-CoV-2 may persist longer in the gastrointestinal tract in cases of innate or adaptive immune dysfunction [86]. Another 16S rRNA sequencing study involving COVID-19 and MIS-C patients reported increased abundances of *Clostridium*, *Dialister*, *Veillonella*, *Ruminococcus*, and *Streptococcus* and decreased levels of *Granulicatella*, *Prevotella*, *Bifidobacterium*, and *Blautia* in MIS-C patients. These findings, although not establishing a direct causative link between gut microbiota and MIS-C development, underscore a potential role of dysbiosis and zonulin-mediated intestinal permeability in the pathophysiology of MIS-C, warranting further investigation [87]. In addition to the role of microbiota composition and barrier integrity in exacerbating inflammation, the NLRP3 inflammasome, which regulates pro-inflammatory cytokines, is also implicated. Its role in MIS-C warrants comprehensive investigation [88]. It is worth noting, however, that many studies examining bacterial flora in KD and MIC-S differ in their methodologies, and factors such as diet, comorbidities, and antibiotic therapy have not always been fully controlled and standardized for all of the factors mentioned. Therefore, it is not possible to establish a direct relationship between these factors without improved standardization of studies in subsequent trials conducted by researchers. It is also important to standardize studies in terms of methodology. Our work cited both 16S rRNA sequencing and metagenomic analysis, although they differed in sequencing depth and the bioinformatics tools used. These differences may affect the identified taxa and the degree of their recognition, which can make it difficult to distinguish direct differences between studies.

### 4.4. Clinical Manifestation

The case definition of the disease includes the presence of characteristic clinical symptoms, elevated inflammatory markers (with at least two laboratory parameters above the normal range), exclusion of alternative microbial causes other than COVID-19, and evidence of SARS-CoV-2 infection. This evidence may include a positive PCR test for SARS-CoV-2 or serological confirmation—i.e., the presence of IgG and/or IgM antibodies against SARS-CoV-2 within 10 days of hospital admission—or documented exposure to individuals infected with the virus [75,89].

The diagnostic criteria include the presence of fever (with no minimum duration required) and CRP ≥ 3 mg/dL, as well as the involvement of at least two organ systems. For the cardiovascular system, symptoms may include shock, myocardial involvement, reduced ejection fraction (≤55%), coronary artery abnormalities, and elevated troponin levels. Mucocutaneous manifestations, such as rashes, are commonly observed; however, unilateral lymphadenopathy is typically absent. Gastrointestinal symptoms include abdominal pain, diarrhea, and vomiting. Hematological findings may show thrombocytopenia (<150,000 cells/μL) and lymphopenia (<1000 cells/μL). The patient should be younger than 21 years of age. A positive PCR test for SARS-CoV-2, positive antigen test, positive serological response (IgG/IgM), or documented exposure to a confirmed or probable COVID-19 case within 60 days prior to symptom onset further supports the diagnosis [90].

Additional clinical manifestations of this syndrome may include myocarditis, while cough and sore throat are reported less frequently [80,91,92]. In a study conducted in New York, 99 patients were admitted with suspected MIS-C, of whom 95 were confirmed cases. These patients presented with a variety of symptoms, including hypotension or shock, cardiac involvement, rash, conjunctivitis, gastrointestinal manifestations, and mucocutaneous signs, thereby meeting the required diagnostic criteria [89]. Upon admission, 63% of patients had a fever, with a median temperature of 38.3 °C. Among cardiovascular symptoms, 97% exhibited tachycardia, 78% tachypnea, and 32% hypotension. Follow-up echocardiography is recommended even in asymptomatic patients, due to the risk of developing new coronary aneurysms during the convalescent phase. In symptomatic cases—such as those with coronary artery dilation or aneurysms—echocardiography should be repeated every 2–3 days. Additionally, a follow-up scan at 4–6 weeks post-diagnosis is advised for all patients [93]. It is important to note that clinical presentation varies widely, with some patients exhibiting signs of shock, while others may present only with inflammation or fever. Shorter time intervals between symptom onset and hospital admission have been associated with worse prognosis [94].

Laboratory findings are predominantly characterized by neutrophilia (83%), elevated CRP levels (94%), and lymphopenia, which was present in 50% of the examined patients. Elevated troponin T levels were observed in 68% of cases, while 77% exhibited increased pro-BNP levels. Echocardiographic abnormalities were identified in 59% of patients, whereas radiological imaging revealed pathological changes in 41% of cases [80]. Additionally, elevated levels of ferritin and IL-6 have been reported, with authors noting a more pronounced inflammatory response compared to KD. Some patients present with thrombocytopenia or normal platelet counts, which contrasts with the thrombocytosis commonly observed in KD [77]. Increased concentrations of chemokine ligand CXCL9, IL-10, IL-17, IL-18, and soluble IL-2 receptor have also been documented [95]. Other studies report elevated erythrocyte sedimentation rates, hypoalbuminemia, increased alanine aminotransferase levels, anemia, elevated D-dimer levels, prolonged INR, and increased fibrinogen concentrations. In addition to the aforementioned parameters, alterations in related biomarkers have also been identified [82].

In general, the clinical manifestations resemble those observed in the aforementioned KD, and treatment protocols similarly involve the administration of IVIG, corticosteroids, and, in more severe cases, biologic agents [92]. Documented therapeutic approaches include the use of infliximab as well as IL-6 and IL-1 antagonists [80]. A study evaluating treatment strategies for MIS-C found no significant difference in recovery outcomes among patients receiving IVIG alone, IVIG combined with corticosteroids, or corticosteroids alone as first-line therapy [96]. However, another study reports that the combined use of IVIG and corticosteroids, compared to IVIG monotherapy, is associated with a reduced risk of developing new or persistent cardiovascular abnormalities [97]. Patients presenting with cardiac dysfunction and accompanying hypotension may require inotropic and vasoactive support [75]. The therapeutic effect of IVIG is mediated through the inhibition of IL-1 and IL-6 cytokine production via Fcγ receptor binding by IgG. Moreover, IVIG administration has been shown to enhance left ventricular systolic function and attenuate disease severity. Corticosteroids suppress cytokine-induced immune responses, reduce mortality, improve cardiac recovery, and shorten intensive care unit stays; however, they also impair immune reactivity, which may be disadvantageous. Aspirin is administered at a dose of 30–100 mg/kg/day, particularly in patients with cardiac involvement. In addition to the above-mentioned therapies, anakinra (an IL-1 receptor antagonist) and tocilizumab (an IL-6 receptor antagonist) are employed. Remdesivir, an RNA polymerase inhibitor, has also been used in selected cases. In patients at risk of thromboembolic complications, antithrombotic or antiplatelet therapy, including enoxaparin and aspirin, is sometimes indicated [98]. Farooqi et al. report that although a significant proportion (76%) of patients present in critical condition, symptoms tend to resolve relatively rapidly [95]. Early administration of immunomodulatory therapy has shown favorable outcomes, including zero mortality, improved left ventricular function, and prevention of myocardial inflammation and fibrosis [99]. The comparison of KD and PIMS/MIS-C is presented in Table 1.

## 5. Discussion

Motivated by the clinical significance and potential therapeutic implications of the microbiome in KD and MIS-C, we aim to systematize the current scientific understanding on this topic. The similarity in clinical presentation between KD and MIS-C is an undisputed fact [100]. Our focus is not on reiterating well-established distinctions, such as the younger age of onset in KD or the tendency toward a more severe systemic inflammatory response in MIS-C [101]. Instead, we strive to critically analyze the most recent and relevant literature to identify both similarities and differences in the potential etiologies and mechanisms in these two conditions.

The current literature indicates that the etiology of KD remains incompletely understood. Genetic, environmental, and infectious factors have been implicated, with the immune response playing a central role. It is worth examining the NLRP3 inflammasome, a central component of the inflammatory response, whose activation may depend on the composition of the gut microbiota and its metabolites [66]. This mechanism has been well described in the context of KD. In MIS-C, to the best of our knowledge, direct data remain limited and require further in-depth validation. According to the current literature, no approved therapeutic agents specifically targeting this structure exist to date; however, research is ongoing into NLRP3 activation inhibitors, with the hope of potential therapeutic applications [102]. According to Tao et al., a key catalytic factor may be the degree of intestinal barrier permeability [4]. In the case of MIS-C, it is similarly proposed that autoimmune mechanisms may play a significant role [103]. The acute phase of the immune response in KD is characterized by exaggerated, uncontrolled innate immune activation, alongside a Th17 response and suppression of substantial fractions of T and B lymphocytes. This is typically triggered by exposure to a foreign antigen. Interestingly, similar processes are observed in MIS-C, where virus-induced cell death leads to the release of damage-associated molecular patterns, which subsequently activate the innate immune system and result in clinical manifestations akin to KD [104]. There are hypotheses suggesting that prolonged persistence of SARS-CoV-2 in the gastrointestinal tract in MIS-C patients may increase intestinal permeability, facilitating the translocation of viral particles into systemic circulation. Within the framework of these potential immunological mechanisms, emphasis is placed on innate, adaptive, and humoral immunity. The literature also highlights the possible presence of autoantibodies targeting host tissues [105]. The study by Yadaw et al. identified markers such as NKp46+ cells, CD11+ eosinophils, Tim-3+ cytotoxic T lymphocytes, and D-dimer levels as significant predictors of severe MIS-C [106]. Hufnagel et al. reported higher levels of D-dimers, acute-phase proteins, ferritin, creatinine, troponin T, NT-proBNP, and neutrophils in MIS-C patients compared to those with KD [107]. Recent findings suggest that *vimentin* and *chloride intracellular channel 1* genes may serve as potential diagnostic biomarkers for KD [108]. Kuo has hypothesized that the antibody profile against *Escherichia coli*, in combination with cytokine levels, could help differentiate KD from other febrile illnesses. Key cytokines associated with KD include IL-6, IL-12, TNF-α, IFN-γ, CXCL10, as well as anti-inflammatory cytokines such as IL-4, IL-5, IL-13, and IL-31 [109]. In contrast, MIS-C is associated with elevated levels of IL-1α, IL-2, IL-6, IL-10, IL-15, IL-17A, IL-33, IL-1Ra, GM-CSF, TNF-α, IFN-α, IFN-β, and IFN-γ [110]. A comprehensive review of the literature suggests that both KD and MIS-C are characterized by heightened immune system activity and systemic inflammation, with MIS-C often presenting with an intense cytokine storm [111]. Currently, no specific laboratory markers have been identified that would allow for unequivocal diagnosis. Beyond cytokine profiles and inflammatory markers, attention is being drawn to novel potential diagnostic indicators, particularly in KD. For instance, Zeng et al. suggest that the oral overgrowth of *Streptococcus* and *Escherichia-Shigella* species may serve as microbial biomarkers for KD [112]. Conversely, Franchitti et al. report that gut microbes such as *Bacteroides* spp., *Bifidobacterium* spp., and *Akkermansia muciniphila* do not correlate with MIS-C severity and thus lack utility as disease progression markers [103]. Suskun et al. propose that decreased levels of *Faecalibacterium prausnitzii* may indicate ongoing gastrointestinal inflammation in MIS-C, though the finding is not disease-specific. The role of the gut microbiome in both KD and MIS-C has garnered considerable attention. Evidence suggests that alterations in microbial composition are present in both conditions. In KD, increased intestinal abundance of *Enterococcus*, *Helicobacter*, *Staphylococcus*, *Bacteroidetes*, and *Acinetobacter* has been noted, with *Enterococcus* and *Helicobacter* particularly linked to elevated IL-6 levels. A reduction in *Firmicutes* has also been reported—similar to findings in MIS-C, where increased prevalence of *Bacteroidetes*, *Clostridium*, *Prevotella*, and *Eggerthella* has been documented compared to healthy or SARS-CoV-2-positive children. Whether these microbial shifts are a cause or consequence of disease remains uncertain. In the case of KD, Wang et al., utilizing a two-sample Mendelian randomization analysis, report no evidence of a genetic association between the disease and the gut microbiota. They emphasize the importance of accounting for confounding factors in future research [113]. In the case of MIS-C, knowledge in this area remains limited and warrants further investigation. Microbiota modifications are clinically relevant, as they may influence therapeutic efficacy. For example, in KD, IVIG treatment may be less effective in the presence of *Enterococcus faecalis* overgrowth and reduced *Bacteroides thetaiotaomicron*, particularly when antibiotic resistance genes are present. In contrast, current data on the direct impact of gut dysbiosis on treatment outcomes in MIS-C are limited. Both acute KD and MIS-C are associated with a depletion of SCFA-producing bacteria, a finding of growing interest given SCFAs’ role in maintaining mucosal immunity and anti-inflammatory homeostasis [6,11,112,114]. The aforementioned factors are known to play a role in coordinating the differentiation of Th17 and Tregs, thereby completing a theoretical feedback loop suggesting a potential link between impaired immune responses and the pathogenesis of KD [115]. Jena et al. demonstrated an association between vasculitis induced by *Lactobacillus casei* cell wall extract—a well-established murine model for KD—and the composition of the intestinal microbiota. They proposed that supplementation with SCFAs could alleviate this inflammatory state by strengthening the intestinal barrier [116]. Similarly, a 2023 study employing probiotics to indirectly enhance SCFA production in a mouse model confirmed the beneficial impact of SCFAs on intestinal barrier stability and showed a marked reduction in inflammation [117]. In an earlier study, Kinumaki et al. emphasized that due to potential intestinal wall dysfunction in KD, interventions targeting the gut microbiota—and thereby the integrity of the intestinal barrier—may offer a preventative or therapeutic advantage by reducing disease susceptibility and mitigating clinical outcomes [68]. Supplementation with SCFA-enhancing agents has been proposed in the context of various immune-mediated disorders. SCFAs, which are well known for their systemic immunomodulatory properties and their critical role in maintaining intestinal barrier integrity [118,119], appear theoretically promising as adjunctive therapy in MIS-C. However, the literature to date does not report any specific clinical studies addressing their use in this context. We hypothesize that this gap in evidence may be attributed to factors such as the relative rarity of MIS-C and the severity of its clinical course, which complicate the design and implementation of targeted interventional trials [120]. KD is not without long-term consequences in terms of gut microbiota alterations. Teramoto et al. suggested that children with a history of KD may exhibit immune-related abnormalities associated with a reduced capacity to sustain colonization by beneficial bacterial microorganisms. The study recruited patients from a single region in Japan, which may constitute a significant limitation. Countries and continents with diverse cultures are characterized by distinct environmental factors, diets, and other variables, which may pose a barrier to the generalization of the findings on a global scale [7]. The existing literature on long-term microbiota changes is primarily focused on post-COVID-19 transformations [121], while the specific context of MIS-C remains insufficiently explored and has yet to be comprehensively described in current scientific research.

Many researchers emphasize the need for a deeper understanding of the underlying mechanisms of the conditions discussed—an assertion with which we fully concur. We recognize ongoing scientific progress in areas that offer promising avenues for enhanced diagnostic and therapeutic support. This includes investigations into gut microbiota alterations and their immunological implications, often utilizing experimental models, biomarker identification, and proposals for adjunctive therapeutic strategies. At the same time, we observe notable gaps in the current body of knowledge—most importantly, the need for methodological standardization. For instance, studies exploring gut microbiota changes in KD are currently limited by confounding factors such as the use of antibiotics and specific dietary interventions, which may obscure results. The aforementioned antibiotics are administered to the majority of patients prior to diagnosis, which substantially impacts the gut microbiome and complicates subsequent interpretation. A significant proportion of studies focus on 16S rRNA analysis, which may present certain limitations due to its resolution constraints. Additional limitations include small sample sizes and population heterogeneity [6,67]. Given that cesarean delivery and formula feeding have been identified as potential risk factors for KD, we argue for the establishment of registries to monitor at-risk individuals and for comparative studies between such cohorts and the healthy population [7]. The review revealed the existence of studies involving both ill and healthy children. Consequently, we also recognize the need to establish registries encompassing children with the disease as well as those with other febrile infections, in order to differentiate specific aspects between these groups. To our knowledge, research concerning MIS-C remains less developed than the extensively studied KD. In the context of MIS-C, further investigation is warranted to assess the direct impact of dysbiosis on treatment efficacy, as well as the potential therapeutic value of microbiota-targeting interventions. We advocate for the implementation of prospective registries aimed at evaluating potential long-term shifts in gut microbial composition. A key research direction should focus on broad microbiological interventions and their influence on disease course, which could ultimately translate into tangible benefits in clinical practice. The summary of gut microbiota changes connected with pathogenesis of described diseases was presented in Figure 1.

## 6. Conclusions

Recent years have seen advances in the understanding of gut dysbiosis in MIS-C and KD.SCFAs hold promise as adjunctive therapy, particularly in KD; however, this aspect remains underexplored in the context of MIS-C.Greater emphasis should be placed on scientific investigations into diagnostic and monitoring biomarkers.The analysis of long-term alterations in gut microbiota is essential due to potential clinical implications. Research on this topic in the context of MIS-C remains significantly limited.

## Figures and Tables

**Figure 1 biomedicines-13-01953-f001:**
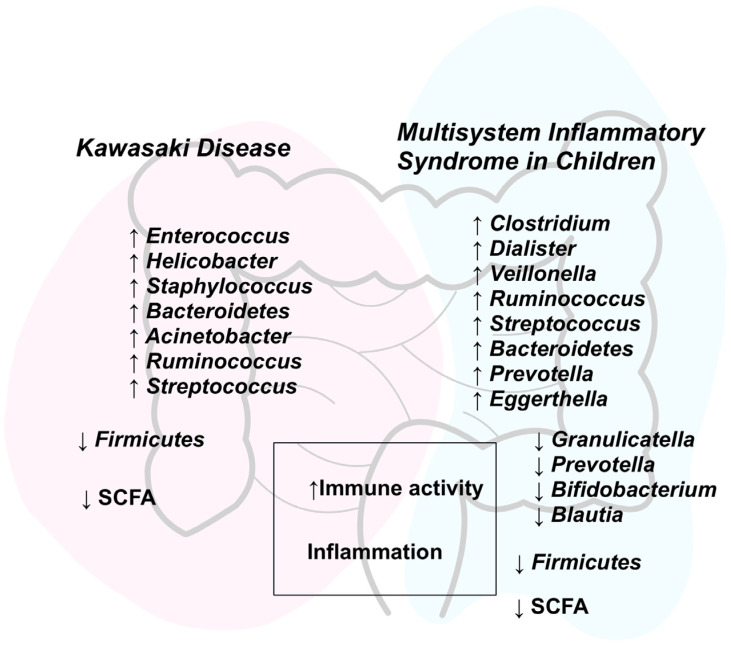
Summary of the review based on the main text and references: bacterial and immunological alterations in KD and MIS-C/PIMS. Abbreviations: SCFA—short chain fatty acids, ↑—up-regulation, ↓—down-regulation.

**Table 1 biomedicines-13-01953-t001:** Comparative characteristics of KD and PIMS/MIS-C in children.

	KD	PIMS/MIS-C
Epidemiology	0.01% of children, mostly aged <5 y.o. [47,48,49].	0.002% of people, mostly aged <21 y.o. with a mean age of 8 y.o. [81].
Trigger	A genetic predisposition and post-infectious influence have been suggested [48].	Associated with SARS-CoV2 infection occurring typically 4–6 weeks after [76].
Main clinical features	Fever, rash, conjunctivitis, cervical lymphadenopathy, and swelling or erythema of the hands and feet [47].	Fever, mucocutaneous manifestations, diarrhea, abdominal pain, and shock [82].
Gut dysbiosis	A decrease in SCFA-producing bacteria and an increase in proinflammatory bacterial species [7,67].	It plays a significant role in disruption of the intestinal barrier and modulating immune responses [86].
Cardiac abnormalities	Coronary artery aneurysms, mitral valve regurgitation, pericardial effusion, myocarditis, and impaired left ventricular function [57,70,71].	Reduced ejection fraction, myocarditis, and cardiogenic shock [80,91,92].
Laboratory abnormalities	Elevated CRP, thrombocytosis, anemia, hypoalbuminemia, elevated transaminase levels, neutrophilia [70,71].	Elevated CRP, neutrophilia, lymphopenia, increased troponin T levels, elevated pro-BNP, elevated D-dimer levels, thrombocytopenia [80].
Treatment	IVIG + asprin, corticosteroids, and/or biologic agents [70].	IVIG, corticosteroids, and biologic agents [92].

## Data Availability

The original contributions presented in this study are included in this article. Further inquiries can be directed to the corresponding authors.

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
