# Peer review of "Intestinal Dysbiosis and Immune Activation in Kawasaki Disease and Multisystem Inflammatory Syndrome in Children: A Comparative Review of Mechanisms and Clinical Manifestations"

_biomedicines, 2025, doi:10.3390/biomedicines13081953_

Round 1

Reviewer 1 Report

Comments and Suggestions for Authors

The review article provides a comprehensive synthesis of existing research on the gut microbiome's role in Kawasaki Disease (KD) and Multisystem Inflammatory Syndrome in Children (MIS-C).

Issues

  • The review discusses microbiome alterations in KD and MIS-C but does not critically evaluate the variability in methodologies across studies (e.g., 16S rRNA sequencing vs. metagenomics, differences in sequencing depth, bioinformatics pipelines). Inconsistent techniques may lead to discrepancies in reported microbial taxa, making it difficult to compare findings across studies.
  • The authors recognize that factors such as antibiotic use, diet, and comorbidities influence the composition of the microbiome; however, they do not systematically evaluate how these confounding variables are controlled in the studies they cite. Without proper adjustment for these factors, the reported changes in microbial composition may indicate confounding effects rather than genuine associations with disease.
  • All abbreviations must be defined in the paper the first time they are used.
  • In lines 205, 244, 251, and 519, the sentences are repetitive or convey the same meaning.
  • Names of all bacterial taxa (kingdoms, phyla, classes, orders, families, genera, species, and subspecies) should be italicized in the manuscript.
  • It’s important to use consistent abbreviations when referring to the same clinical entity. For clarity, you could explain that both "MIS-C" and "PIMS" refer to the same condition, but you will choose one abbreviation throughout the text to avoid confusion for the readers.
  • The NLRP3 inflammasome could be incorporated into this text as a potential mechanistic link between intestinal dysbiosis and immune activation in both entities linking dysbiosis to Inflammation via NLRP3. This could be added to: Section 3.3 (Intestinal Dysbiosis in KD) and Section 4.3 (Intestinal Dysbiosis in MIS-C). In the Discussion, you could highlight NLRP3 as a unifying mechanism and potential therapeutic target.

Author Response

Comment 1: The review discusses microbiome alterations in KD and MIS-C but does not critically evaluate the variability in methodologies across studies (e.g., 16S rRNA sequencing vs. metagenomics, differences in sequencing depth, bioinformatics pipelines). Inconsistent techniques may lead to discrepancies in reported microbial taxa, making it difficult to compare findings across studies.

Response 1: Thank you very much for this valuable comment. We fully agree that methodological differences between studies can significantly influence microbiome results. We have carefully revised the manuscript to address this issue and included a more critical reflection on how variations in sequencing techniques and analysis pipelines may affect the comparability of findings.

Comment 2: The authors recognize that factors such as antibiotic use, diet, and comorbidities influence the composition of the microbiome; however, they do not systematically evaluate how these confounding variables are controlled in the studies they cite. Without proper adjustment for these factors, the reported changes in microbial composition may indicate confounding effects rather than genuine associations with disease.

Response 2: We appreciate this insightful suggestion. In response, we have revised the manuscript to more clearly acknowledge the potential impact of confounding factors such as antibiotics, diet, and comorbidities. We have taken care to incorporate this consideration into our discussion of the findings and their limitations.

Comment 3: All abbreviations must be defined in the paper the first time they are used.

Response 3: Thank you for your comment. We have carefully reviewed the manuscript and ensured that all abbreviations are defined at their first occurrence. Any missing definitions have now been added to improve clarity for the reader.

Comment 4: In lines 205, 244, 251, and 519, the sentences are repetitive or convey the same meaning.

Response 4: We appreciate your attention to clarity and conciseness. The mentioned lines have been revised to eliminate repetition and redundancy. The content has been deleted where necessary to avoid conveying the same meaning multiple times while retaining the essential information.

Comment 5: Names of all bacterial taxa (kingdoms, phyla, classes, orders, families, genera, species, and subspecies) should be italicized in the manuscript.

Response 5: Thank you for this important observation. We have reviewed the manuscript and ensured that all bacterial taxa names at all taxonomic levels are now correctly italicized according to the appropriate scientific conventions.

Comment 6: It’s important to use consistent abbreviations when referring to the same clinical entity. For clarity, you could explain that both "MIS-C" and "PIMS" refer to the same condition, but you will choose one abbreviation throughout the text to avoid confusion for the readers.

Response 6: Thank you for your suggestion. We agree that consistent terminology enhances clarity. To maintain consistency and minimize confusion, we have chosen to use the abbreviation “MIS-C” throughout the manuscript.

Comment 7: The NLRP3 inflammasome could be incorporated into this text as a potential mechanistic link between intestinal dysbiosis and immune activation in both entities linking dysbiosis to Inflammation via NLRP3. This could be added to: Section 3.3 (Intestinal Dysbiosis in KD) and Section 4.3 (Intestinal Dysbiosis in MIS-C). In the Discussion, you could highlight NLRP3 as a unifying mechanism and potential therapeutic target.

Response 7: Thank you for this valuable and insightful suggestion. In response, we have incorporated relevant content into Section 3.3 and Section 4.3, discussing the role of the NLRP3 inflammasome in promoting inflammation in the context of dysbiosis. Additionally, we have revised the Discussion section to highlight NLRP3 as a potential unifying inflammatory pathway in both conditions and as a possible therapeutic target. These additions help to strengthen the mechanistic understanding and translational relevance of the findings.

Reviewer 2 Report

Comments and Suggestions for Authors

Thank you for the opportunity to review your manuscript.

Your work is comprehensive and well-structured. 

I think the following suggestions could improve the strength of the manuscript.

Abstract is extensive may be beneficial to shorten and focus on the main conclusions and implications.

Discussion:

More critical analysis of the literature would be beneficial. How strong are the evidence  linking specific microbial changes to disease mechanisms or outcomes?Are there any limitations in current studies related for example with geographic, methodological, or population data?

Table and Figure:

    • Table 1 is helpful, adding references or footnotes would enhance clarity

    • Figure 1 should be better explained in the main text

Comments on the Quality of English Language

  • Consider refining long or complex sentences to improve clarity and flow.
  • Avoid repetition in the introduction and discussion for better readability.

Author Response

Comment 1: Abstract is extensive may be beneficial to shorten and focus on the main conclusions and implications.

Response 1: Thank you for the suggestion. We have revised the abstract to make it more concise and focused, emphasizing the main conclusions and their implications while removing less essential details to improve readability.

Comment 2: More critical analysis of the literature would be beneficial. How strong are the evidence  linking specific microbial changes to disease mechanisms or outcomes?Are there any limitations in current studies related for example with geographic, methodological, or population data?

Response 2: We appreciate this insightful comment. We have expanded the Discussion section to include a more critical evaluation of the literature, addressing the strength of the evidence linking microbial changes to disease mechanisms and outcomes. Additionally, we highlighted current study limitations, including geographic variability, methodological differences, and population heterogeneity, to provide a balanced perspective.

Comment 3: Table 1 is helpful, adding references or footnotes would enhance clarity.

Response 3: Thank you for this helpful suggestion. We have added relevant references to Table 1 to improve clarity and allow readers to better understand the source of the data presented.

Comment 4: Figure 1 should be better explained in the main text.

Response 4: We agree that Figure 1 would benefit from a clearer explanation. Additionally, we would like to clarify that Table 1 was created based on the information presented in the main text, and its title has been updated accordingly to better reflect its content.

Round 2

Reviewer 1 Report

Comments and Suggestions for Authors

Dear Authors,

I am pleased to inform you that I have reviewed the revised version of your manuscript. I appreciate the careful attention you have given to my previous comments and suggestions. Your revisions have improved the manuscript, addressing all the points raised during the review process. The clarifications, additional methodological details, and expanded discussions have strengthened the paper's scientific rigor and overall quality.

I approve your manuscript for publication in its current form.